# Trends in the Volume and Types of Primary Care Visits during the Two Years of the COVID-19 Pandemic in Israel

**DOI:** 10.3390/ijerph191710601

**Published:** 2022-08-25

**Authors:** Oren Miron, Yael Wolff Sagy, Shlomit Yaron, Noga Ramot, Gil Lavie

**Affiliations:** 1Branch of Planning and Strategy, Clalit Health Services, Tel Aviv 62098, Israel; 2Health Policy and Management, Ben-Gurion University of the Negev, Beer Sheva 84105, Israel; 3Community Medical Services Division, Clalit Health Services, Tel Aviv 62098, Israel; 4Ruth and Bruce Rappaport Faculty of Medicine, Technion-Israel Institute of Technology, Haifa 31096, Israel

**Keywords:** COVID-19 pandemic, primary care visits, virtual care, telehealth, low value care

## Abstract

Background: The outbreak of the COVID-19 pandemic led to a decrease in primary health care in-person visits and a simultaneous increase in virtual encounters. Objective: To quantify the change in the total volume of primary care visits and mix of visit types during the two years of the pandemic in Israel. Design: Cross-sectional study. Participants: All primary care visits by members of the largest healthcare organization in Israel, during three one-year periods: the pre-COVID-19 year (March 2019–February 2020), the first year of COVID-19 (March 2020–February 2021), and the second year of COVID-19 (March 2021–February 2022). Main measures: Total volume of primary care visits and mix of visit types. Results: More than 112 million primary care visits were included in the study. The total visit rate per 1000 members did not change significantly between the pre-COVID year (19) and the first COVID year (19.8), but was 21% higher in the second COVID-19 year (23). The rate of in-person visits per 1000 members decreased from 12.0 in the pre-COVID year to 7.7 in the first COVID year and then increased to 9.6 in the second. The rate of phone visits and asynchronous communication increased from 0.7 and 6.3, respectively, in the pre-COVID year, to 4.1 and 8, respectively, in the first COVID year, and remained unchanged in the second. There was substantial variation across age groups and sectors in the adoption of virtual platforms. Conclusions: The rapid introduction of virtual encounters in primary care tended to displace in-person visits in the first year of the pandemic, but they appear to have been additive in the second. This transition should be monitored, with the goal of ensuring appropriate planning efforts and resource allocation to deal with the potential added burden on medical staff. Efforts should be invested in encouraging the use of virtual platforms in patient groups that currently underutilize it, such as minorities.

## 1. Introduction

The COVID-19 pandemic dramatically altered patterns of health care delivery during the first year of the pandemic [1], leading to an increase in the difficulty and risk associated with in-person visits, which, in turn, led to a decrease in emergency department visits [2], myocardial infarction admissions [3], and also a decrease in ambulatory in-person visits and an increase in virtual encounters [1,4]. This latter transition was not associated with increased overall visit rates in primary care during the first year of the pandemic [5,6], suggesting that virtual care tended to replace in-person visits rather than having an additive effect, and provided a vital avenue for care delivery [5]. It is unclear whether this trend was maintained during the pandemic and how it affected the total volume of visits.

Previous studies have shown that younger age groups tend to adopt virtual care at a faster rate and minorities at a slower one [7,8], with potential exacerbations of health disparities. Yet, it is unclear whether changes in the volume and mix of visits persisted over time among patients in various age groups and sectors.

The current study attempts to determine whether there has been a long-term increase in the volume of visits and change in the mix of visit types within primary care during the two years of the COVID-19 pandemic in Israel, and if so, whether it is dependent on age or sector. Such an increase is liable to result in a greater burden on healthcare workers who experienced unprecedented strain during the pandemic. The healthcare system, therefore, needs to undertake the necessary modifications to deal with this potential added burden [9].

## 2. Methods

The study’s dataset included all primary care visits by members of Clalit Health Services (CHS), which insures 4.8 million members (52% of the Israeli population). The visits were divided according to three time periods of twelve consecutive months each, with the goal of reducing the effect of seasonal variation: pre-COVID (March 2019–February 2020), the first COVID-19 year (March 2020–February 2021) and the second COVID-19 year (March 2021–February 2022). Note that most of the Israeli adult population had received the COVID-19 vaccine by early in the third period.

For each month in each period, the rate of average daily visits per 1000 CHS members was calculated. The average rate was then calculated for each of the three periods. A comparison was also carried out by type of visit: in-person, phone (which includes video calls, though they only accounted for one percent of all virtual care visits), and asynchronous communication (by means of a mobile app or website, which permits offline communication with the physician, including requests for a drug prescription and laboratory tests).

Due to the divergence of visit rates during the two months of the Omicron wave (at the end of the third period, i.e., January–February 2022), the results are also broken down according to the first ten and last two months of each period.

The analysis was performed for all patients and also by age group (0–18, 19–44, 45–64 years, 65+) and by sector (general Jewish, ultra-Orthodox Jewish and Arabs).

Socioeconomic status (SES) was based on the small statistical areas (SSA) used in the 2008 Israeli census. SSAs contain 3000–4000 persons and are created to maintain homogeneity in terms of the sociodemographic composition of the population. The Israeli Central Bureau of Statistics utilizes demography, education, employment, housing conditions, and household income to define the socio-economic level of the population in each SSA (Israeli Central Bureau of Statistics: Jerusalem, Israel) [10]. POINTS^©^ Location Intelligence Company continuously improves the accuracy of the SES measure, using up-to-date sociodemographic, commercial, and real estate data [11], and classifies them into ten categories, from 1 (lowest) to 10 (highest).The comparisons were carried out using one-way analysis of variance (ANOVA) together with the Tukey honest significant difference (HSD) test between each pair of time periods. The statistical significance level was set at *p* < 0.05, and the analysis was performed with R version 4.0.1 (R core team, Vienna, Austria).

## 3. Results

More than 112 million primary care visits were identified (mean age of 43; 56% women) which took place over a three-year period. The rate of primary care visits was higher among women, the elderly, high-SES individuals and the general Jewish sector (Table 1).

During the first year of the pandemic, there was a decline of 35% in the in-person visit rate (−4.25 visits per 1000 persons, 95% CI (−5.45, −3.06)), which occurred primarily during the early months of the pandemic (Figure 1). This was accompanied by an increase of 496% in the phone visit rate (+3.43 visits, 95% CI (2.14, 4.73)) and an increase of 26% in asynchronous communication (+1.69 communications, 95% CI (0.58, 2.80)) (Table 2). It appeared that virtual encounters usually replaced in-person visits, with the overall visit rate during the first COVID year (19.8) remaining basically unchanged relative to the pre-pandemic rate (19) (Figure 1 and Figure 2a).

Relative to the first year of the pandemic, the in-person visit rate (9.58) increased by 24% during the second year of the pandemic (+1.866 visits, 95% CI (0.675, 3.057)), although it was still lower than the pre-pandemic rate (11.97). The change in the rate of phone visits and asynchronous communication between the first and second year of the pandemic was not statistically significant: +0.44 visits, 95% CI (−0.85, 1.74) and +0.76 visits, 95% CI (−0.33, 1.87), respectively. Relative to the pre-pandemic period, there was an overall increase of 21% in the total visit rate (+3.95 visits, 95% CI (1.14, 6.76)), which was most pronounced during the Omicron wave (Table 2 and Appendix A, Figure 1 and Figure 2a).

In view of the fact that the Omicron wave led to a massive surge in COVID-19 cases (Appendix A) and, in turn, a substantial increase in phone visits for the purpose of monitoring individuals who had been infected, we divided the third period into the first ten months of the year and the two months of the Omicron wave (January–February 2022), and compared them to the corresponding months in the pre-COVID year and the first COVID year.

The comparisons showed that relative to the corresponding pre-pandemic months, the rate of in-person visits per 1000 members in January–February 2022 decreased by 28% (−3.679 visits, CI (−4.250, −3.109)) while the phone visit rate increased by 7.918 (95% CI (3.275, 12.561)). Overall, visits during the Omicron period increased by 7.897 (95% CI (−1.441, 17.234)) per 1000 members, which represented a 37% increase relative to the corresponding pre-pandemic months, although it was not statistically significant due to the short time period and monthly level of analysis.

Relative to the equivalent months in the pre-pandemic year, the in-person visit rate decreased by 18% in March–December 2021 (−2.135 visits, 95% CI (−3.533, −0.738)); the phone visit rate increased by 454% (+3.079 visits, 95% CI (2.475, 3.684)); and the asynchronous communication rate increased by 36% (+2.225 visits, 95% CI (1.199, 3.251)). This led to an overall increase of 17% in the total visit rate (+3.169 visits, 95% CI (0.702, 5.637)) compared with the relevant ten months in the pre-pandemic year (Appendix A).

### 3.1. Comparison by Age Group (Table 3)

#### 3.1.1. Overall Visits

The total visit rate for the pre-COVID year increased with age groups. The visit rate for the 0–18 age group declined between the pre-COVID period and the first COVID year by 13% (−1.93 visits, 95% CI (−4.74, 0.87)), while it increased by 13% in the 19–44 age group (2.13 visits, 95% CI (−0.80, 5.07)), by +9% in the 45–64 age group (2.22 visits, 95% CI (−0.93, 5.38)) and by 9% in the 65+ age group (3.12 visits, 95% CI (0.29, 5.94)). Between the first COVID year and the second COVID year, the total visit rate in the latter groups increased by 42% (5.1 visits, 95% CI (2.29, 7.90)), 13% (2.41 visits, 95% CI (−0.52, 5.35)), 7% (1.87 visits, 95% CI (−1.27, 5.03)) and 2% (0.9 visits, 95% CI (−1.92, 3.73)), respectively. Between the pre-COVID year and the second COVID year, the total visit rate increased by 22.7% in the 0–18 age group (+3.16 visits, 95% CI (0.35, 5.97)) and by 28.4% in the 19–44 age group (+4.55 visits, 95% CI (1.61, 7.49)), while it increased by 17.5% in the 45–64 age group (4.1 visits, 95% CI (0.94, 7.26)) and by 12.1% in the 65+ age group (4.02 visits, 95% CI (1.20, 6.85)).

**Table 3 ijerph-19-10601-t003:** ANOVA of visits by age.

Age Group	Type of Visit	Yearly Mean Daily Visit Rates per 1000 Persons			Differences between Pairs of Years					
		Pre-COVID	COVID Year 1	COVID Year 2	Pre-COVID to COVID Year 1		Pre-COVID to COVID Year 2		COVID Year 1 to COVID Year 2	
					Diff (95% CI)	*p*-Value	Diff (95% CI)	*p*-Value	Diff (95% CI)	*p*-Value
0–18 years-old	In-person visits	10.651	5.718	8.840	−4.93 (−6.45, −3.40)	0.000	−1.81 (−3.33, −0.28)	0.017	3.12 (1.59, 4.64)	0.000
	Phone visits	0.317	2.736	3.761	2.41 (1.04, 3.79)	0.000	3.44 (2.07, 4.81)	0.000	1.02 (−0.34, 2.39)	0.174
	Asynchronous	2.971	3.551	4.504	0.58 (−0.28, 1.44)	0.243	1.53 (0.66, 2.40)	0.000	0.95 (0.08, 1.82)	0.029
	Total visits	13.939	12.005	17.105	−1.93 (−4.74, 0.87)	0.224	3.16 (0.35, 5.97)	0.024	5.10 (2.29, 7.90)	0.000
19–44 years old	In-person visits	9.459	6.436	7.407	−3.02 (−3.85, −2.18)	0.000	−2.05 (−2.88, −1.21)	0.000	0.97 (0.13, 1.80)	0.020
	Phone visits	0.686	3.864	4.285	3.17 (1.82, 4.52)	0.000	3.59 (2.24, 4.94)	0.000	0.42 (−0.92, 1.77)	0.727
	Asynchronous	5.827	7.805	8.832	1.97 (0.66, 3.28)	0.002	3.00 (1.69, 4.31)	0.000	+1.02 (−0.28, 2.33)	0.147
	Total visits	15.973	18.105	20.524	2.13 (−0.80, 5.07)	0.192	4.55 (1.61, 7.49)	0.002	2.41 (−0.52, 5.35)	0.124
45–64 years old	In-person visits	13.906	9.989	11.046	−3.91 (−5.10, −2.72)	0.000	−2.86 (−4.05, −1.66)	0.000	1.05 (−0.13, 2.24)	0.089
	Phone visits	0.957	4.908	5.087	3.95 (2.61, 5.28)	0.000	4.12 (2.79, 5.46)	0.000	0.17 (−1.15, 1.51)	0.943
	Asynchronous	8.534	10.726	11.367	2.19 (0.87, 3.51)	0.000	2.83 (1.51, 4.15)	0.000	0.64 (−0.67, 1.96)	0.465
	Total visits	23.397	25.622	27.500	2.22 (−0.93, 5.38)	0.210	4.10 (0.94, 7.26)	0.009	1.87 (−1.27, 5.03)	0.323
65 years-old or older	In-person visits	18.808	12.859	14.734	−5.94 (−7.64, −4.25)	0.000	−4.07 (−5.76, −2.38)	0.000	1.87 (0.18, 3.56)	0.027
	Phone visits	1.281	7.240	6.445	5.95 (4.72, 7.19)	0.000	5.16 (3.92, 6.40)	0.000	−0.79 (−2.03, 0.44)	0.270
	Asynchronous	12.907	16.190	15.845	3.11 (1.83, 4.39)	0.000	2.93 (1.66, 4.21)	0.000	−0.17 (−1.45, 1.10)	0.940
	Total visits	32.996	36.117	37.024	3.12 (0.29, 5.94)	0.028	4.02 (1.20, 6.85)	0.004	0.90 (−1.92, 3.73)	0.714

#### 3.1.2. Type of Visit

In-person visit rates declined between the pre-COVID period and the first COVID year by 46% in the 0–18 age group (−4.93 visits, 95% CI (−6.45, −3.40)), by 32% in the 19–44 age group (−3.02 visits, 95% CI (−3.85, −2.18)), by 28% in the 45–64 age group (−3.91 visits, 95% CI (−5.10, −2.72)) and by 32% in the 65+ age group (−5.94 visits, 95% CI (−7.64, −4.2 5)); and then increased in the second year of the pandemic by 54% (3.12 visits, 95% CI (1.59, 4.64)), 15% (0.97 visits, 95% CI (0.13, 1.80)), 10% (1.05 visits, 95% CI (−0.13, 2.24)) and 14% (1.87 visits, 95% CI (0.18, 3.56)), respectively. Though an increase was noticed between the first and the second year of the pandemic, among all age group members the rate of in-person visits in the second COVID year was still lower compared with the pre-COVID year.

Phone visit rates increased between the pre-COVID period and the first COVID year by 760% in the 0–18 age group (2.41 visits, 95% CI (1.04, 3.79)), by 462% in the 19–44 age group (3.17 visits, 95% CI (1.82, 4.52)), by 412% in the 45–64 age group (3.95 visits, 95% CI (2.61, 5.28)) and by 464% in the 65+ age groups (5.95 visits, 95% CI (4.72, 7.19)). Phone visits continued to increase in the second COVID year in the 0–18 age group (by 37%, +1.02 visits, 95% CI (−0.34, 2.39)), but did not change significantly in any of the older groups between the first and the second COVID years. 

Phone visit rates increased with age groups during pre-COVID and also after two pandemic years. Though the rate of increase in adoption of this avenue was lower in older age groups during the pandemic, their pre-pandemic preference was significantly higher, as was their absolute rate during pandemic years compared with younger age groups.

Asynchronous communication rates increased between the pre-COVID period and the first COVID year by 19% in the 0–18 age group (0.58 visits, 95% CI (−0.28, 1.44)), by 33% in the 19–44 age group (1.97 visits, 95% CI (0.66, 3.28)), by 25% in the 45–64 age group (2.19 visits, 95% CI (0.87, 3.51)) and by 25% in the 65+ group (3.11 visits, 95% CI (1.83, 4.39)). Asynchronous communication rates continued to increase in the second COVID year in the 0–18 age group (by 26%, +0.95 visits, 95% CI (0.08, 1.82)), thus, resulting in a 55% increase over the two years of the pandemic, but did not change significantly in any of the older groups between the first and the second COVID years (Table 3). 

Similar to phone visits, asynchronous communication rates increased with age groups during pre-COVID and also after two pandemic years. Similarly, although the rate of increase in adoption of this avenue was lower in older age groups during the pandemic, their pre-pandemic preference was significantly higher, as was their absolute rate during pandemic years, compared with younger age groups.

### 3.2. Comparison by Sector (Table 4, Figure 2b)

#### 3.2.1. Overall Visits

The total visit rate for the pre-COVID year was higher among the general Jewish sector (20.953) than among the ultra-Orthodox Jewish (16.98) and Arab (14.56) sectors. During the first year of the pandemic, overall visit rates remained relatively unchanged in all three sectors. However, relative to the pre-pandemic year, there was a large increase (24%) in visits during the second year of the pandemic in the general Jewish sector (+4.89 visits, 95% CI (1.65, 8.14)), compared with a more modest 19% increase in the ultra-Orthodox Jewish sector (+3.27 visits, 95% CI (0.10, 6.43)), and a statistically insignificant increase of 14% increase in the Arab sector (+2.04 visits, 95% CI (−0.10, 4.18)).

**Table 4 ijerph-19-10601-t004:** ANOVA of visits by sector.

Sector	Type of Visit	Yearly Mean Daily Visits Rates per 1000 Persons			Differences Between Pairs of Years					
		Pre-COVID	COVID Year 1	COVID Year 2	Pre-COVID to COVID Year 1		Pre-COVID to COVID Year 2		COVID Year 1 to COVID Year 2	
					Diff (95% CI)	*p*-Value	Diff (95% CI)	*p*-Value	Diff (95% CI)	*p*-Value
General	In-person visits	11.891	7.153	8.915	−4.738 (−5.971, −3.506)	0.000	−2.976 (−4.208, −1.743)	0.000	1.763 (0.530, 2.995)	0.004
	phone visits	0.789	5.018	5.520	4.230 (2.755, 5.704)	0.000	4.731 (3.257, 6.206)	0.000	0.502 (−0.973, 1.976)	0.684
	Asynchronous	8.274	10.352	11.411	2.079 (0.666, 3.492)	0.003	3.137 (1.724, 4.550)	0.000	1.058 (2.471, 0.173)	0.173
	Total visits	20.953	22.523	25.846	1.570 (−1.672, 4.813)	0.468	4.893 (1.650, 8.135)	0.002	3.323 (0.080, 6.565)	0.044
Ultra-Orthodox Jewish	In-person visits	10.562	6.103	8.360	−4.458 (−5.786, −3.128)	0.000	−2.202 (−3.532, −0.872)	0.000	2.256 (0.926, 3.586)	0.000
	Phone visits	0.732	4.955	4.369	4.222 (2.920, 5.525)	0.000	3.637 (2.334, 4.939)	0.000	−0.586 (−1.888, 0.717)	0.519
	Asynchronous	5.683	7.780	7.516	2.096 (0.673, 3.519)	0.003	1.833 (0.410, 3.256)	0.009	−0.264 (−1.687, 1.159)	0.893
	Total visits	16.978	18.838	20.245	1.186 (−1.305, 5.025)	0.331	3.267 (0.102, 6.432)	0.042	1.407 (−1.758, 4.572)	0.526
Arab	In-person visits	12.478	9.500	11.571	−2.977 (−4.321, −1.634)	0.000	−0.907 (−2.250, 0.436)	0.237	2.070 (0.724, 3.413)	0.002
	Phone visits	0.441	1.800	2.370	1.359 (0.375, 2.343)	0.005	1.929 (0.945, 2.913)	0.000	0.570 (−0.414, 1.554)	0.342
	Asynchronous	1.643	2.346	2.657	0.703 (0.259, 1.146)	0.001	1.014 (0.571, 1.458)	0.000	0.312 (−0.132, 0.755)	0.211
	Total visits	14.562	13.646	16.598	−0.916 (−3.054, 1.223)	0.551	2.036 (−0.102, 4.175)	0.065	2.952 (0.813, 5.901)	0.005

#### 3.2.2. Type of Visit

In-person visit rates declined significantly between the pre-COVID period and the first COVID year in all sectors, with a modest increase in the second COVID year. In-person visit rates were still significantly lower after two pandemic years in general Jewish and ultra-Orthodox Jewish sectors (−2.976, 95% CI (−4.208, −1.743), and −2.202, 95% CI (−3.532, −0.872), respectively), but did not change significantly after two pandemic years in the Arab sector compared with the pre-pandemic year (−0.907, 95% CI (−2.250, 0.436)) (Figure 2b, Table 4).

Phone visit rates increased significantly between the pre-COVID period and the first COVID year in all three sectors, demonstrating a large increase in general Jewish and ultra-Orthodox Jewish sectors (4.230, 95% CI (2.755, 5.704), and 4.222, 95% CI (2.920, 5.525), respectively), with only a modest increase in the Arab sector (1.359, 95% CI (0.375, 2.343)). No significant change was noticed during the second pandemic year in all three sectors, compared with the first COVID year.

The asynchronous communication rate was significantly lower prior to the pandemic in the Arab sector (1.6) compared with the general Jewish sector (8.3) and the ultra-Orthodox Jewish sector (5.7). During the first year of the pandemic, there was an increase of 25% (2.079 visits, 95% CI (0.666, 3.492)) in the general Jewish sector, 35% (2.096 visits, 95% CI (0.673, 3.519)) in the ultra-Orthodox Jewish sector, and 42% (0.703 visits, 95% CI (0.259, 1.146)) in the Arab sector. In the second year of the pandemic, the use of asynchronous communication continued to increase in the general Jewish sector (by 1.1 visits relative to the first year of the pandemic, 95% CI (0.173, 2.471)), while there were no significant changes in the other two sectors (Figure 2b, Table 4). 

Overall, during the 2 years, there was a 25% decline in the rate of in-person visits in the general Jewish sector (−2.976 visits, 95% CI (−4.208, −1.743)); a 600% increase in telephone visits (+4.731 visits, 95% CI (3.257, 6.206)); and a 38% increase in asynchronous communication (+3.13 visits, 95% CI (1.724, 4.550)). This compared to a decline in in-person visits of about 20% in the ultra-Orthodox Jewish sector (−2.2 visits, 95% CI (−3.532, −0.872)); a 500% increase in telephone visits (+3.637 visits, 95% CI (2.334, 4.939)); and an increase of 32% in asynchronous communication (+1.833 visits, 95% CI (0.410, 3.256)). 

In the Arab sector, there was no significant change in the in-person visit rates during the two years of the pandemic relative to the pre-COVID year. There was a 430% increase in telephone visits (+1.929 visits, 95% CI (0.945, 2.913)) and a 62% increase in asynchronous communication (+1.014 visits, 95% CI (0.571, 1.458)), although the absolute adoption rates of the latter two avenues of communication were significantly lower for the Arab sector compared with the other two sectors.

## 4. Discussion

In this nationwide sample of members of the largest healthcare organization in Israel, we observed no significant changes in the overall volume of primary care visits during the first year of the pandemic relative to the pre-COVID year, with the growth in virtual care visits being offset by the decline in in-person visits. During the second year, there was a 21% increase relative to the pre-COVID year (a 17% increase when excluding the Omicron period). Phone visits increased sharply during the first COVID year, alongside a significant increase in asynchronous communication. Both remained relatively unchanged during the second pandemic year, but the rate of in-person visits, which had decreased during the first year, increased from the first to the second year (though it remained lower than during the pre-COVID year). In both the first and second pandemic years, virtual care visits surpassed in-person visits.

The first COVID year was characterized by a high level of uncertainty and a significant decline in the consumption of many healthcare services, to the point of concern about possible public health consequences [3]. During the year, there was a step increase in the assimilation of virtual services which tended to replace in-person visits [1,12]. During the second year of the pandemic, when most of the Israeli adult population had already been vaccinated, a recovery was identified in the pattern of healthcare service utilization, which returned to the vicinity of pre-pandemic levels [2]. Therefore, the second year of the pandemic may be viewed as a better predictor of future patterns of healthcare service use, perhaps reflecting a “new normal” reality of living side-by-side with the pandemic.

The use of virtual encounters became more prevalent during the first year of the pandemic and remained at a similar level in the second. Therefore, it appears that virtual care visits became an integral part of the interaction between patients and their physicians, offering an additional and highly convenient platform for obtaining medical services [13,14]. The number of in-person visits declined significantly during the first year of the pandemic but recovered during the second, thus, leading to an increase in the overall number of visits. This increase is liable to increase the burden on healthcare staff. Therefore, healthcare organizations and public health officials should plan a response to this contingency, allocate the necessary resources and make the modifications required by this new reality [15], with the goal of maintaining a high level of medical service.

With respect to age and sector, total visits during the two years of the pandemic grew the most among younger age groups (0–18 and 19–44) and in the general Jewish sector. The rate of adoption of virtual platforms was also the highest in these groups, despite the fact that they tend to be healthier [16] (with the exception of the high increase in asynchronous visits in the Arab sector, limited by low absolute usage rate). This trend may widen health gaps and lead to the inefficient use of resources by healthier groups at the expense of less healthy ones. Unlike older patient groups, characterized by lower rates of increase in virtual adoption but with high baseline pre-pandemic levels of usage, minority sectors (mainly the Arab sector) demonstrated a low virtual adoption rate together with low absolute rates before the pandemic. The variation by age and sector confirms the need to improve adoption of virtual care among minority sectors, patient groups which are currently characterized by underutilization [17,18,19].

This study had several strengths. It examined the effect of the pandemic on primary care utilization patterns in a large integrative healthcare organization over a period of three complete years. Further advantages of the study include the extent of population coverage and the availability of computerized up-to-date data gathered, thus, enabling the monitoring of patients over time.

This study had several limitations. First, the reasons for the medical encounters were not included in the analysis, therefore, it was not possible to determine the clinical conditions that required an in-person encounter following a virtual one. Future research is needed in order to categorize encounters according to clinical diagnoses, with the goal of identifying medical scenarios in which the virtual encounter is a genuine substitute for the in-person encounter and, therefore, not of low-value [10,20].

Second, the study looked at the total number of patient–physician visits in primary care, without relation to the average duration of each type of encounter. This should be taken into account in future research with the goal of better assessing the burden on medical staff.

Third, we only used a one year period as a pre-COVID period, which might not represent the “whole” pre-COVID period.

Fourth, the study was limited to primary care visits in Israel and the results may not be generalizable to other countries.

## 5. Conclusions

Among members of the largest healthcare organization in Israel, the rapid introduction of virtual care visits in primary care tended to displace in-person visits in the first year of the pandemic, but they appeared to have been additive in the second. This transition may reflect a “new normal” pattern in the utilization of healthcare services and should be continuously monitored to ensure appropriate planning efforts and resource allocation that are needed to prepare for the potential added burden on medical staff. Finally, efforts should be invested to encourage the use of virtual platforms in marginalized patient groups who currently underutilize it, such as minorities.

## Figures and Tables

**Figure 1 ijerph-19-10601-f001:**
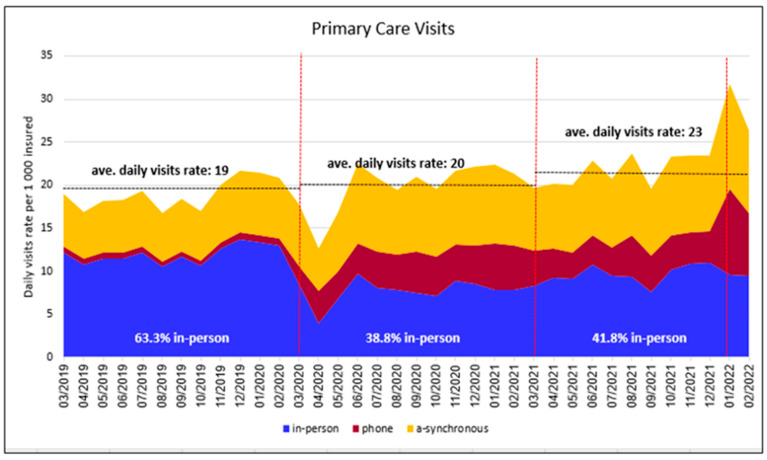
Primary care visits by type. Y axis: primary care visits per 1000 members. X axis: the calendar month of the visit. Blue area represents in-person visits; red area represents phone visits; and yellow area represents asynchronous communication. The dotted horizontal lines show the average visit rate in each of the three study periods. The first red dotted vertical line indicates the start of the pandemic in Israel (March 2020), the second dotted line indicates the start of the COVID-19 vaccine protective period for the majority of Israelis (March 2021), and the third dotted line indicates the start of the Omicron wave (January 2022).

**Figure 2 ijerph-19-10601-f002:**
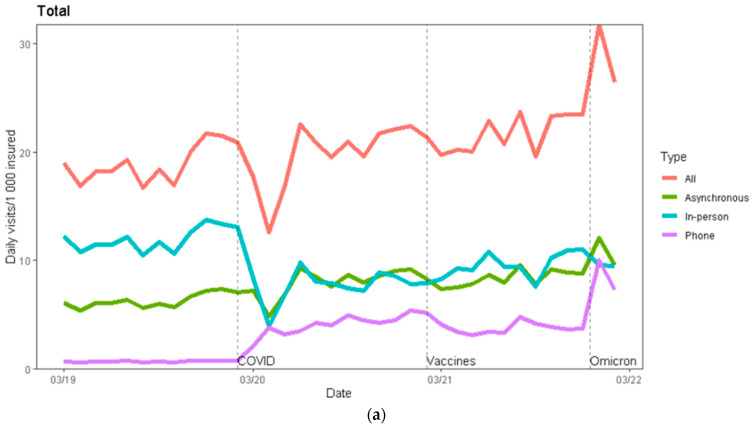
(**a**). Total visits by type. Y axis: primary care visits per 1000 members. X axis: the calendar month of the visit. The red line represents overall visits; the green line represents asynchronous communication; the blue line represents in-person visits; and the purple line represents phone visits. The dotted vertical lines and names indicate (from left to right) the start of the COVID-19 pandemic in Israel (March 2020), the start of the COVID-19 vaccine protective period for the majority of Israelis (March 2021), and the start of the Omicron wave (January 2022). (**b**). In-person, phone, asynchronous and total visits, by sector. Y axis: primary care visits per 1000 members in that sector. X axis: the calendar month of the visit. The red line indicates overall visits; the green line indicates asynchronous communication; the blue line indicates in-person visits; and the purple line indicates phone visits. The dotted vertical lines indicate (from left to right) the start of the COVID-19 pandemic in Israel (March 2020), the start of the COVID-19 vaccine protective period for the majority of Israelis (March 2021), and the start of the Omicron wave (January 2022).

**Table 1 ijerph-19-10601-t001:** Demographics characteristics.

	General Population	Total Visits (N)	In-Person Visits	Phone Visits	Asynchronous Visits
**Total N**	4.76 M	112,086,370	54,014,735	16,503,773	41,567,862
**Gender (% female)**	51%	56% (63,008,751)	54% (29,337,327)	59% (9,698,117)	58% (23,973,307)
**Age (mean SD)**	33	43	38 (26.048)	42 (26.009)	45 (24.904)
**Age Groups**					
0–18 years	35%	24% (27,300,466)	30% (16,270,197)	25% (4,127,399)	17% (6,902,870)
19–44 years	34%	30% (33,406,096)	27% (14,501,298)	32% (5,279,657)	33% (13,625,141)
45–64 years	18%	22% (24,398,247)	21% (11,291,679)	21% (3,394,357)	23% (9,712,211)
65+ years	14%	24% (26,981,561)	22% (11,951,561)	22% (3,702,360)	27% (11,327,640)
**SES**					
Low (1–3)	27%	18% (20,428,648)	24% (13,093,659)	14% (2,327,331)	12% (5,007,658)
Medium–low (4–5)	27%	28% (31,911,721)	29% (15,468,805)	28% (4,687,080)	28% (11,755,836)
Medium–high (6–7)	27%	31% (34,276,897)	26% (13,821,503)	33% (5,514,915)	36% (14,940,479)
High (8–10)	17%	18% (19,934,612)	14% (7,752,610)	20% (3,283,569)	21% (8,898,433)
Missing	2%	5% (5,534,492)	7% (3,878,158)	4% (690,878)	2% (965,456)
**Sector**					
General Jewish	66%	75% (83,546,243)	64% (34,413,474)	80% (13,187,053)	86% (35,945,716)
Ultra-Orthodox Jewish	7%	6% (6,384,187)	5% (2,890,166)	7% (1,118,301)	6% (2,375,720)
Arabs	27%	20% (21,992,252)	31% (16,595,056)	13% (2,188,794)	8% (3,208,402)
Missing observations	0%	0% (163,688)	0% (116,039)	0% (9625)	0% (38,024)

**Table 2 ijerph-19-10601-t002:** Primary care visits by type and year.

Type of Visit	Mean Daily Visit Rates per 1000 Patients			Differences between the Periods					
	Pre-COVID Year	First COVID Year	Second COVID Year	Pre-COVID and First COVID Year		Pre-COVID and Second COVID Year		First COVID Year and Second COVID Year	
				Diff (95% CI)	*p*-Value	Diff (95% CI)	*p*-Value	Diff (95% CI)	*p*-Value
In-person visit	11.972	7.714	9.580	−4.25 (−5.45, −3.06)	0.000	−2.39 (−3.58, −1.20)	0.000	1.866 (0.675, 3.057)	0.001
Phone visit	0.692	4.131	4.578	3.43 (2.14, 4.73)	0.000	3.88 (2.58, 5.18)	0.000	0.44 (−0.85, 1.74)	0.678
Asynchronous communication	6.305	7.999	8.769	1.69 (0.58, 2.80)	0.002	2.46 (1.35, 3.57)	0.000	0.76 (−0.33, 1.87)	0.217
Total visits	18.969	19.844	22.926	0.87 (−1.93, 3.68)	0.728	3.95 (1.14, 6.76)	0.004	3.08 (0.27, 5.89)	0.029

## Data Availability

The data are not publicly available due to privacy restrictions.

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
