# Peer review of "Trends in the Volume and Types of Primary Care Visits during the Two Years of the COVID-19 Pandemic in Israel"

_ijerph, 2022, doi:10.3390/ijerph191710601_

Round 1

Reviewer 1 Report

This nation-base study analyses trends in the primary health-care during the COVID-19 pandemic. It focuses mainly on the type of communication and visits in the primary sector during the period.

The study is well conducted and presents data for half of the Israeli population. This is also the strongest point of the study. The language is clear; the results are well presented and properly analyzed in the discussion sector.

The weak point of the study is the lack of description of reasons for different types of visits during the pandemic years. It does not describe weather there were changes in the spectrum of diseases and weather there were changes in the quality of health-care when the types of communications changed.

Still, the study is interesting, analyses a large population and the results are worth publishing since they bring important insight regarding COVID-19 and primary health-care. Therefore I would recommend the study to be published.

Author Response

All of the reviewer's comments were answered in the updated manuscript

Reviewer 2 Report

Dear Authors,

Thank you for your work on this issue. Overall, this version of the manuscript is well-written. The authors described the trends in volumes and types of primary care visits during the pandemic. Some comments are provided below to strengthen the value of this manuscript.

1. The introduction can improve by adding some relevant references regarding the impact of the pandemic on the COVID pandemic, for example, how the pandemic affects the healthcare system (ED visits, complications of diseases, etc.)

2. Some acronyms were not defined (i.e., SES (I believe it stands for socioeconomic status), POINTS, etc.)

3. Figures 1 and 2a are somewhat similar. Please consider editing these figures or combining them.

4. If possible, please consider adding the chief complaint (or diagnosis of primary care visits) and comparing them between 3 periods. It would add some consequences to this pandemic.

5. Please consider telling the readers about the current situation of the "telemedicine" or "phone visit" in Israel.

6. Since the authors used only one period (1 year) as a pre-COVID period, it might not represent the "whole" pre-COVID period, and the effects after this period may not be concluded. It might be just an incidental effect and needs to be stated as one of the limitations. 

Author Response

All of the Reviewer's comments were answered in the updated manuscript

Round 2

Reviewer 2 Report

Thank you the authors for revising this manuscript. This version is much improved and I think it's worthy of publication. Well done!

Author Response

Minor changes were made